# Enhancing Regulatory T Cells to Treat Inflammatory and Autoimmune Diseases

**DOI:** 10.3390/ijms24097797

**Published:** 2023-04-25

**Authors:** Tara Fiyouzi, Hector F. Pelaez-Prestel, Raquel Reyes-Manzanas, Esther M. Lafuente, Pedro A. Reche

**Affiliations:** Laboratory of Immunomedicine, Faculty of Medicine, University Complutense of Madrid, Ave Complutense S/N, 28040 Madrid, Spain

**Keywords:** regulatory T cells, mucosal tolerance, mucosal immunity, autoimmune disease, adoptive Treg cell transfer, microbiota, vitamin D3, SCFA, inflammatory bowel disease

## Abstract

Regulatory T cells (Tregs) control immune responses and are essential to maintain immune homeostasis and self-tolerance. Hence, it is no coincidence that autoimmune and chronic inflammatory disorders are associated with defects in Tregs. These diseases have currently no cure and are treated with palliative drugs such as immunosuppressant and immunomodulatory agents. Thereby, there is a great interest in developing medical interventions against these diseases based on enhancing Treg cell function and numbers. Here, we give an overview of Treg cell ontogeny and function, paying particular attention to mucosal Tregs. We review some notable approaches to enhance immunomodulation by Tregs with therapeutic purposes including adoptive Treg cell transfer therapy and discuss relevant clinical trials for inflammatory bowel disease. We next introduce ways to expand mucosal Tregs in vivo using microbiota and dietary products that have been the focus of clinical trials in various autoimmune and chronic-inflammatory diseases.

## 1. Introduction

Immune tolerance is a characteristic of the immune system designed to avoid and contain immune responses that could damage the host [1]. Self-tolerance and tolerance to innocuous environmental antigens prevent autoimmune, allergic and chronic-inflammatory diseases. Immune tolerance is particularly relevant at the oral and gut mucosa, as they are highly exposed to many environmental antigens, including those from food and the microbiota. The mucosa is also the entry gate for many pathogens, and therefore, the balance between immune defensive responses and immune tolerance must be tightly regulated [2].

Immune tolerance is maintained via two main mechanisms: central tolerance and peripheral tolerance [1]. Central tolerance is acquired owing to the negative selection during B and T lymphocyte development in primary lymphoid organs (thymus and bone marrow). This process eliminates any lymphocyte that strongly recognizes self-antigens. B cell precursors recognizing extracellular self-antigens in neighboring cells are the main targets of negative selection [3]. In the case of T cells, antigen-recognition requires the participation of antigen presenting cells (APCs) and negative selection applies also to intracellular antigens. During T cell development in the thymus, dendritic cells (DCs) and medullary thymic epithelial cells undertake this task, presenting peptides derived from intracellular and extracellular self-antigens, bound to major complex histocompatibility (MHC) molecules. Subsequently, developing T lymphocytes (thymocytes) expressing T cell receptors (TCRs) with strong affinity for self-peptide–MHC complexes are eliminated [4]. Incidentally, thymocytes must also recognize peptide–MHC complexes with little or moderate affinity to complete their maturation [4].

The process of central tolerance does not eliminate all potentially self-reactive lymphocytes, mainly because lymphocyte progenitors are not exposed to all potential autoantigens during development. Moreover, T cells are particularly prone to self-reactivity since they must also recognize self-peptide–MHC complexes to survive thymic selection. Hence, there is a need for secondary mechanisms of tolerance in the periphery [1]. There are several mechanisms of peripheral immune tolerance, including the acquisition of anergy by auto-reactive lymphocytes. Anergy is a state of unresponsiveness developed by circulating lymphocytes after recognizing self-antigens in the absence of co-stimulation [5]. Another main mechanism of peripheral immune tolerance involves the participation of immunosuppressive cells.

There is a large variety of cells with immunosuppressive activity, including Tregs [6,7], regulatory B cells [8], regulatory NK cells [9], tolerogenic DCs (TolDCs) [10], immunosuppressive macrophages [11], and epithelial [12,13] and endothelial cells [14]. Arguably, Tregs are the most ubiquitous and important cells guaranteeing peripheral immune tolerance and homeostasis [6,7]; Treg cells are essential not only to control misguided immune responses and to maintain self-tolerance but also to avoid excessive immune reactions [15]. Incidentally, defects in Treg cell function and/or numbers have been described in most autoimmune diseases (e.g., systemic lupus erythematosus, rheumatoid arthritis, type 1 diabetes mellitus, multiple sclerosis, myasthenia gravis, etc.) and chronic-inflammatory related disorders [16]. Hence, enhancing regulatory T cell function has an enormous therapeutic potential [15,17,18].

In this review, we will discuss fundamental aspects of Treg cell biology, including their ontogeny, subtypes and activity. We will pay particular attention to the biology and function of mucosal Tregs, as they are essential to avoid harmful immune reactions prompted by environmental agents such as the microbiota and food antigens [19]. In addition, we will discuss relevant interventions and treatments to enhance Treg cell numbers and function that have been tested in clinical trials for autoimmune and chronic-inflammatory diseases, including allergy.

## 2. Regulatory T Cells: Ontogeny and Mechanisms of Action

Tregs were originally described as suppressor T cells for their ability to suppress auto-reactive immune responses. However, their name was later changed to regulatory T cells as they generally control immune responses [20]. Although some Tregs are CD8^+^ [21] most regulatory T cells are CD4^+^, and by default, Tregs are considered CD4^+^ T cells. There are several subtypes of Tregs, but the most important and well-understood group expresses the master transcriptional factor forkhead box P3 (FoxP3), concomitantly with high levels of the CD25 cell surface marker [22]. FoxP3^+^ Tregs can be generated during T cell development in the thymus and are subsequently known as thymus-derived Tregs (tTregs) or naturally occurring Tregs (nTregs) [6,23]. Specifically, tTregs derive from strongly self-reactive CD4^+^ T cell precursors that are rescued from negative selection by DCs under the influence of thymic stromal lymphopoietin (TSLP) produced by the Hassall’s corpuscles [24]. Mature tTregs can represent up to 5–10% of all circulating CD4^+^ T cells and are distributed throughout all lymphoid tissues [25]. FoxP3^+^ Tregs can also be generated in the periphery from circulating CD4^+^ FoxP3^−^ T cells recognizing self-antigens under sub-immunogenic conditions. These Tregs are known as peripheral Tregs (pTreg) [26]. Naive CD4^+^ T cells are often pointed as the source of pTregs but other sources are also plausible. In fact, the preferential source of pTregs in mice appears to be recent thymic emigrants (RTEs) with an intrinsic propensity to acquire a FoxP3^+^CD25^+^ Treg phenotype rather than mature naive CD4^+^ T cells. These RTE have an enhanced sensitivity to FoxP3 inducing factors such as IL-6, TCR triggering and TGF-β, which is absent in naive CD4^+^ T cells [27]. tTregs can be distinguished from pTregs by the expression of markers such as programmed cell death-1 (PD-1), CD73, neuropilin 1 (Nrp1), and Helios, which appears to be higher in tTregs [28]. However, both tTregs and pTregs co-exist in the same sites and the differences between them are indeed too subtle to set them apart [29]. Moreover, both, tTregs and pTregs, can develop from T cells with identical TCR specificity, as shown in OVA-TCR transgenic OTII mice [30]. 

Tregs can also be generated in vitro by stimulating naive CD4^+^ T cells with TGF-β and IL-2 to induce FoxP3 expression. The resulting Tregs are known as induced Treg (iTregs) [31,32]. Interestingly, iTregs can be generated with defined antigen-specificity, which is highly relevant for the development of Treg-based interventions for autoimmune diseases with known autoantigens [33]. Likewise, it is also possible to promote the differentiation of Th1, Th2 and Th17 into FoxP3^+^ Treg cells in vitro [34,35,36]. The suppressive activity of iTregs is similar to that of tTregs and pTregs, but they are more likely to lose it, concomitantly with FoxP3 expression [31,32]. tTregs are the most stable FoxP3^+^ Tregs but under certain conditions, they can convert into conventional T cells (Tcons), losing their suppressive activity [29,37]. The plasticity of FoxP3^+^ Tregs is likely due to the existence of Treg populations with variable degrees of lineage commitment, although some intrinsic flexibility to adapt their phenotype to the environment cannot be ruled out [38,39]. Understanding the factors that influence the plasticity of FoxP3+ Tregs is important for their therapeutical applications.

Regulatory T cells lacking FoxP3 expression have also been identified. These FoxP3^−^ Tregs include Type 1 regulatory T cells (Tr1) [40] and T helper 3 cells (Th3) [41]. Tr1 cells produce large amounts of IL-10 and are characterized by their unique cytokine expression profile: IL-10, TGF-β, IFN-γ, IL-5, IL-4 and IL-22 [40]. On the other hand, Th3 cells primarily secrete TGF-β and can be recognized by their low expression of CD25 and moderate levels of Glucocorticoid-induced TNFR-related protein (GITR) and Cytotoxic T-Lymphocyte Antigen 4 (CTLA-4) [41]. Owing to the production of TGF-β, Th3 cells are important for IgA production in the mucosa [42]. Overall, the biology, function and relevance of Tr1 and Th3 are less understood than that of FoxP3^+^ Tregs. Thereby, we will focus on the immunosuppressive mechanisms of FoxP3^+^ Tregs.

Treg cells require TCR stimulation in the presence of IL-2 to become activated and exert their immunosuppressive action [43,44]. To that end, Tregs must recognize cognate peptide antigens displayed on the cell surface of APCs bound to MHC class II (MHC II) molecules through their TCRs [45]. Once activated, Tregs can suppress the response of many different immune cells, particularly that of Tcons, regardless of antigen-specificity [7,46,47]. Hence, antigen-specific Tregs can induce a state of systemic and bystander immunosuppression to many unrelated antigens [7,48]. Treg cell immunosuppression is mediated via three main mechanisms: (1) contact-dependent suppression, (2) metabolic disruption of Tcon cells, and (3) the secretion of inhibitory cytokines such as IL-10, IL-35 and TFG-β, which inhibit both T cells and DCs [47,48] (Figure 1). Treg cells dampen the stimulatory properties of DCs in a contact-dependent manner by engaging CD80/86 and MHC molecules with inhibitory receptors such as CTLA-4 and LAG-3, respectively [49]. Tregs can thus suppress any T cell, CD4^+^ or CD8^+^, recruited by the same DC [47]. Metabolic disruption of effector T cells by Tregs is mediated by the release of cAMP, the generation of adenosine by Treg ectoenzymes CD39 and CD73 and the consumption of IL-2 [50,51]. In addition, Tregs induce apoptosis on effector T cells through membrane-tethered TGF-β [52] and delivery of granzyme B [53]. Finally, Tregs promote a tolerogenic environment that facilitates the differentiation of more Tregs from Tcons [54].

## 3. Tregs in Mucosal Tissues

Mucosa-associated lymphoid tissues are characterized by an elevated presence of Tregs to maintain tolerance to commensal microbiota and harmless environmental antigens such as those in food [2,19]. Focusing on the gut mucosa, FoxP3^+^ Tregs represent between 20 and 30% of all CD4^+^ T cells and are characterized by the constitutive expression of CTLA-4 and inducible T cell co-stimulator (ICOS), as well as by the secretion of IL-10, TGF-β and IL-35 [55]. Treg subtypes residing in the gut mucosa are summarized in Figure 2. About 20% of those Tregs are GATA3^+^ tTregs, which expand locally owing to the presence of microbial metabolites such as butyrate [56] and retinoic acid (RA) produced by DCs and epithelial cells [57], and are likely cross-reactive with environmental antigens [58]. These Treg cells are particularly active under inflammatory conditions, activating other Tregs and preventing excessive inflammation [55]. The remaining 80% of Tregs are thought to derive from uncommitted CD4^+^ T cells that recognize the microbiota or dietary antigens and differentiate into pTregs with the help of environmental factors such as TGF-β, RA and microbial metabolites including butyrate [59,60,61]. In the gut mucosa, some pTregs express the nuclear receptor and transcription factor RAR-related orphan receptor γt (RORγt). RORγt^+^NRP1^−^FoxP3^+^ pTregs represent ~60% of pTregs and appear to be induced by microbial antigens [62,63,64]. pTregs lacking the expression of RORγt represent ~20% of these cells and seem to be induced by dietary antigens, thus playing an important role in the susceptibility to food allergies [59]. Given the similarity between tTreg and pTreg, it is also plausible that pTregs are cross-reactive with self-antigens.

Resident DCs in mucosal tissues participate in the generation and maintenance of FoxP3^+^ Tregs as well as other subtypes of Tregs through mechanisms that are dependent on IL-27, TGF-β, IL-10, RA, indoleamine-2,3-dioxygenase and vitamin D [65]. Moreover, DCs play an important role in the induction of antigen-specific tolerance. For example, it has been shown that pulmonary DCs can induce suppressive Tr1 cells in an IL-10 and IL-27 dependent manner after the nasal administration of antigens [66,67]. In contrast, the ingestion of specific food antigens has been shown to prompt gut DCs to induce antigen-specific Th3 cells in a manner dependent on TGF-β secretion [67]. Overall, it is important to understand how DCs enhance Tregs to induce antigen-specific tolerance. Next, we will review some approaches to induce Tregs for therapeutic and prophylactic purposes.

## 4. Adoptive Treg Cell Transfer Therapy: An Overview 

Adoptive Treg cell transfer (ATregCT) represents a direct therapeutic approach to suppress aberrant or excessive immune responses [68,69]. ATregCT was first introduced for the treatment and prevention of graft-versus-host-disease (GVHD) after hematopoietic stem cell transplantation [70,71]. There is an increasing interest in its application in autoimmune and chronic-inflammatory diseases, including type 1 diabetes (T1D) [72], systemic lupus erythematosus (SLE) [73,74], multiple sclerosis (MS) [75] and inflammatory bowel disease (IBD) [76]. At the time of this writing, there are over 54 registered clinical trials exploring ATregCT therapies for different diseases characterized by uncontrolled immune responses (ClinicalTrials.gov).

ATregCT therapy involves the isolation of Tregs from patients, followed by an ex vivo expansion of the cells prior to their autologous transference [69,77]. Tregs used in ATregCT are typically isolated from peripheral mononuclear blood cells (PMBCs) using magnetic or fluorescence activated cell sorting with relevant antibody panels [78]. The most recent advances in cell sorting have enabled the isolation of pure populations of T cells with a CD4^+^CD25^+^CD127^low^CD45RA^+^ phenotype, which seems to be the most appropriate population for ATregCT therapy [76,78]. Subsequently, Tregs are expanded in vitro using different protocols aimed to imprint Tregs with enhanced and enduring immunosuppressive properties [77]. A common protocol consists of culturing Tregs with rapamycin and/or retinoic acid followed by the activation with anti-CD3/CD28 antibodies in the presence of IL-2 and/or TGF-β [79,80]. Tregs expanded in this manner are polyclonal and are widely used in clinical trials. However, Tregs used for ATregCT can also be antigen-specific. For this purpose, antigen-specific Tregs are generated by incubating Tregs with the appropriated antigens [81]. Because Tregs require antigen stimulation, antigen-specific Tregs have the advantage of providing local immunosuppression at the site/tissue of antigen encounter. In this regard, it has been shown that alloantigen-specific Tregs are more efficient than polyclonal Tregs in controlling GVHD [82,83]. Autoantigens targeted in autoimmune diseases are prime candidates to generate antigen-specific Tregs for ATregCT. Unfortunately, information on autoantigens is limited or unavailable for many autoimmune diseases, precluding this approach [33,83].

### Adoptive Treg Cell Transfer Therapy for Inflammatory Bowel Disease

The gastrointestinal tract is constantly exposed to environmental stimuli and is highly susceptible to the loss of immune tolerance, leading to chronic disorders such as inflammatory bowel disease (IBD). IBD includes two distinct conditions, ulcerative colitis (UC), affecting the colon mucosa, and Crohn’s disease (CD), which can affect any portion of the intestinal tract. In both UC and CD, inflammation occurs as a result of an inappropriate immune response to resident bacteria [84]. IBD immunopathology is characterized by a local imbalance between FoxP3^+^ Tregs and effector T cells, with Th1 and Th17 cells driving inflammation [85,86]. Moreover, defects in FoxP3^+^ Tregs increases the risk for IBD [87].

Current therapeutic approaches for IBD include anti-inflammatory and immunosuppressant drugs and anti-TNF biologics. Although anti-TNF blockers can control disease progression in some patients, about one-third of the patients do not respond to the treatment or stop responding over time [88]. ATregCT is an alternative and promising therapeutic approach for IBD in these cases. In mouse models of colitis, ATregCT therapy has been shown to control and prevent intestinal inflammation [89]. Subsequently, several distinct ATregCT therapeutic approaches have been tested in human clinical trials for CD and UC (Figure 3). These clinical trials are summarized in Table 1 and will be discussed down below. 

A phase I/IIa clinical trial (Eudract, Number: 2006-004712-44) evaluated the safety and efficacy of a single-injection of ovalbumin-specific Treg cells (ova-Tregs) in patients with refractory CD [90]. In this study, an escalating dose of a single intravenous injection of autologous ova-Tregs (from 10^6^ to 10^9^ cells) was administered to 20 patients. Autologous Tregs expanded from PBMCs were cultured with ovalbumin, IL-2 and IL-4 for 7 days and cloned according to the methods described by Brun et al. [91]. Ova-Tregs were selected based on an ovalbumin-specific IL-10 and Tr1 cytokine production profile. Selected Ova-Tregs were kept in nitrogen tanks and thawed prior to injection into patients [90]. During a 12-week intervention, patients had a normal diet supplemented with meringue cake to ensure Ova-Treg activation at the site of inflammation. Efficacy was assessed using Crohn´s Disease Activity Index (CDAI), IBD Quality-of-Life Questionnaire, C-reactive protein and fecal calprotectin levels. Results indicated that 40% of the patients had a reduction in CDAI levels at weeks 5 and 8, with the best results seeing in the lower dose group (10^6^ cells). The safety profile showed good tolerability regardless of the cell dose. Treatment-emergent adverse events were mostly related to the gastrointestinal system and the underlying CD. As the maximum improvement in patients was observed at week 5, authors concluded that multiple regular injections are required to maintain the efficacy of this therapy. 

In a randomized controlled double-blind phase IIb clinical trial initiated in 2014 (NCT02327221), Ova-Tregs were tested in patients with active CD. In this study, two intravenous doses of Ova-Tregs or placebo were administered for 16 weeks, followed by a third and fourth administration during a period of 16 weeks of the open-label phase. According to the primary outcome measures that were recorded, the 6-week post-administration of 10^6^ cells could induce a strong reduction in CDAI.

In another phase I/II clinical trial starting in 2018 (NCT03185000), autologous polyclonal CD25^+^CD127^low^CD45RA^+^ Tregs (named TR004) were tested for CD. CD45RA^+^ Tregs were chosen in this study because FoxP3 expression is more stable in these cells. Additionally, according to previous preclinical studies, these cells could home to the small bowel when tested in a xenotransplant mouse model with severe combined immune deficiency [76]. Hence, these cells can be recruited to zones of inflammation and locally modulate immune responses. This clinical trial is still ongoing and new updates will be posted on ClinicalTrials.gov.

For UC, a phase I clinical trial was initiated in 2021 (NCT04691232) to test the safety and tolerability of autologous polyclonal Tregs in a study named ER-TREG 01 [92]. Preliminary reported results involve a patient suffering from refractory UC with concomitant primary sclerosing cholangitis with a history of unresponsiveness to conventional therapies. This patient received a single dose of 10^6^ autologous ex vivo expanded Tregs per kg of body weight. Tregs were isolated after leukapheresis, cultured in the presence of IL-2 and rapamycin, and then expanded with anti-CD3/anti-CD28 beads. Clinical and histological signs of improvement were observed and maintained until week 12 of Treg cell therapy. The biopsies taken from the inflammatory zones of the intestine confirmed that the transferred Tregs had migrated to the mucosa [93].

## 5. Prompting the Expansion of Tregs In Vivo for Therapeutic and Prophylactic Purposes

An alternative approach to adoptive Treg cell transfer therapy is to stimulate the patient’s own Tregs in vivo by emulating natural mechanisms. The most suitable sites for promoting the expansion of Tregs are mucosal tissues, particularly the intestinal mucosa, as they harbor inductive mechanisms enabling the differentiation of antigen-specific pTregs that are required to maintain tolerance to environmental agents such as the microbiota and food antigens. Interestingly, naturally induced pTregs at the gut mucosa can also provide systemic bystander immunosuppression by the mechanisms described above (Figure 1).

### 5.1. Induction of Tregs by Gut Microbiota

There is compelling evidence for the influence of the gut microbiota on Tregs, which could be manipulated with prophylactic and therapeutic benefits [64,94]. The gut microbiota can promote the induction of Tregs through the production of Short-Chain Fatty Acids (SCFAs) derived from microbial fermentation of dietary fiber (Figure 4) [95]. In DCs, butyrate, an SCFA, reduces pro-inflammatory gene expression and promotes a tolerogenic phenotype through signaling via the G protein-coupled receptor GPR109A. These tolerogenic DCs (TolDCs) subsequently induce Treg cell differentiation and increases the number of IL-10 producing-T cells [96]. Moreover, in naive T cells, butyrate signaling via the G protein-coupled receptor GPR43 directly induces pTreg differentiation directly by inhibiting histone deacetylase (HDAC) and promoting histone H3 acetylation at the promoter and conserved non-coding sequence regions of the FoxP3 gene [61]. These effects can be reproduced in vivo with HDAC inhibitors, which increase FoxP3 expression and Treg cell expansion and suppressive function [97]. SFCA and dietary fiber (a major source of microbiota-produced SFCA) are being tested in several clinical trials for rheumatoid arthritis and MS (Table 2).

There is also evidence that the composition of the microbiota can affect Treg number and function [98]. Probiotics containing species belonging to the genera *Clostridia*, *Lactobacilli* and *Bifidobacterium*, and *Bacteroides fragilis* can increase the population of Tregs in the colon of germ-free animals. In a study by Narusima et al. [99], animals fed with a mixture of 46 *Clostridia* strains isolated from conventionally reared mice and 17 strains of human-derived *Clostridia* promoted the accumulation of *Clostridia* antigen-specific pTreg cells and enhanced suppressive activity. The human commensal *Bacteroides fragilis* has also been shown to increase Treg cell numbers and IL-10 production in mice, and oral treatment with *B. fragilis* prevents or reduces inflammation and disease activity scores in different experimental models, including lupus nephritis [100] and IBD [101]. Treg induction by *B. fragilis* can be linked to the release of vesicles containing polysaccharide A (PSA) (Figure 4). In fact, the oral administration of purified PSA reproduced the same prophylactic and therapeutic effects of whole bacteria in a mouse model of experimental autoimmune encephalomyelitis, increasing IL-10 secreting B cell and T cell populations [102,103]. PSA binds to TLR1/TLR2 on DCs and induces IL-10 production, which in turn activates Tregs to produce IL-10 [103]. In recent studies, PSA has also been shown to induce the expression of immune checkpoint markers LAG3, TIM3 and PD1, involved in tolerance [104]. In another study using a mouse model of induced allergy, the probiotic mixture VSL#3 composed of 8 strains of *Bifidobacterium*, *Lactobacillus* and *Streptococcus* increased Treg numbers and reduced food allergy inflammation in correlation with augmented TFG-β beta production [105]. *Lactobacillus reuteri* has also been shown to increase Treg cell numbers in the intestinal mucosa and to ameliorate disease scores in a mouse model of rheumatoid arthritis [106].

In summary, the composition of the microbiota can be modulated to boost Treg cell populations in vivo. However, the duration of such treatments is unclear, as the existing microbiota is largely stable over time and quite resistant to modification [107,108]. In Table 2, we summarize the information from various clinical trials evaluating the effect of probiotics on chronic-inflammatory diseases such as atopic dermatitis and allergic rhinitis.

### 5.2. Induction of Tregs with Micronutrients

There are nutrients that can also promote Treg cell expansion and function. Among them, vitamin A and D are the most widely studied [109]. Information from several clinical trials evaluating the effect of vitamin A and D on Treg cells induction in the context of autoimmune diseases is summarized in Table 2. We next discuss the mechanisms by which vitamin A and D promote Treg cell expansion and function.

Dietary vitamin A is metabolized to retinoic acid (RA) by epithelial cells and mucosal DCs that express the enzyme aldehyde dehydrogenase. RA is the active form of vitamin A that promotes the induction of Tregs. Briefly, RA delivered by mucosal DCs is taken up and in combination with TGF-β promotes Treg cell differentiation [110,111]. RA also supports the expansion of the CD161^+^ Tregs, required for wound healing [112] and it has been shown to upregulate the expression of two gut-homing markers on T cells, the chemokine receptor CCR9 and the integrin α4β7, contributing to Treg accumulation in the gut [113]. Vitamin A has been tested in various clinical trials for different diseases, including MS and chronic immune thrombocytopenia among others, as summarized in Table 2.

Vitamin D (VD3) can be absorbed from the diet or produced endogenously in the dermis from 7-dehydroxycholesterol in response to ultraviolet radiation. VD3 is converted to its active form 1,25(OH)2D3 by two sequential hydroxylations that take place in the liver and kidney, respectively, or locally in the cytoplasm of DCs. Active VD3 is transferred during antigen presentation to T cells, where it binds to vitamin D response elements in non-coding regions of the FoxP3 gene, increasing its expression and prompting pTreg differentiation [114]. Low levels of VD3 in serum correlates with the severity of the symptoms in many autoimmune diseases, including type 1 and type 2 diabetes, MS and SLE [115,116,117]. Supplementation with VD3 also results in increased pTreg numbers and IL-10 production in patients with autoimmune and chronic inflammatory diseases, including rheumatoid arthritis, SLE, MS and T1D. Ongoing and past clinical trials are aimed at determining the dose and forms of VD3 supplementation with a positive effect on disease scores (Table 2). Additionally, VD3 has been proposed as an adjuvant in antigen-specific tolerogenic vaccines to favor TolDCs inducing pTregs [118,119,120].

Food antigens can also induce oral tolerance by expanding Treg cells in the small intestine, which suppress immunogenic responses to food. Groundbreaking studies in germ-free mice have shown that animals develop pTregs that are specific of food antigens. These pTregs are distinguishable from microbiota-induced pTreg cells and suppress immune responses to ingested protein antigens [59]. Food-derived peptides, such as those from pepsin-hydrolyzed ovalbumin, can induce TolDCs via TLR signaling and induce Treg cell expansion and IL10 production, reducing Th2 responses [121]. The oral administration of peanut proteins induces a rapid desensitization, as observed in a phase III clinical trial in a European pediatric cohort (ARTEMIS) [122]. In autoimmune diseases, the oral administration of antigens or modified antigens has been used in MS (myelin basic protein), uveitis (retinal antigens), rheumatoid arthritis (type II collagen) and T1D (insulin peptides). These strategies have shown promising results in early stage clinical trials, but need to be confirmed in phase III trials [123,124,125].

### 5.3. Inducing Tregs with Low Dose IL-2 Therapy

Interleukin-2 (IL-2) plays an important role in adaptive immunity, and it is required for homeostasis, proliferation, survival and induction of Foxp3 Tregs [126,127]. The effects of IL-2 are dose-dependent. High-dose IL-2 promotes antitumoral immunity by enhancing the activation of cytotoxic CD8 T cells and NK cells and it is being investigated for the treatment of cancer [128]. On the other hand, low-dose IL-2 leads to the expansion of Tregs, which can be used for the treatment of autoimmune and chronic-inflammatory diseases [129]. This is due to the high levels of CD25 (IL2RA) expressed on the surface of Tregs, which together with the IL-2 receptor subunit beta (IL2RB) and the common gamma chain (IL2RG) form the heterotrimeric IL-2 receptor complex [130]. Low-dose IL-2 therapy has shown to be effective in inducing the expansion of CD4^+^CD25^+^FoxP3^+^ T cells in clinical trials in autoimmune diseases including T1D [131], SLE [132] and RA [133] (Table 2). Data obtained from these studies suggest that a dosage of 10^6^ IU/m^2^ daily during a variable course of therapy is well tolerated and provides satisfactory clinical response in GVHD and autoimmune diseases [134,135]. More recently, IL-2 complexes with specific monoclonal antibodies such as JES6–1 (IL-2/JES6) have been developed to prolong IL-2 half-life and provide a sustained and selective IL-2 signal to CD25^+^ cells. These complexes promote a potent expansion of Tregs while minimizing the side effects of higher doses of IL-2 [136,137].

**Table 2 ijms-24-07797-t002:** Clinical trials in autoimmune and chronic-inflammatory diseases, including allergy, enhancing Treg cells in vivo.

	Study ID	Disease	Treatment	Status	Ref.	Outcomes/Objective
Probiotics	NCT01500941	Atopic Dermatitis	*Lactobacillus salivarius* and *Bifidobacterium breve* (daily)	C	[138]	Clinical parameters improved. Th17/Treg ratio progressively decreased in the probiotic group. Effect remains 2 months after the discontinuation of treatment.
	NCT02349711	Allergic Rhinitis	*L. gasseri*, *B. bifidum*, and *B. longum*) (twice a day)	C	[139]	Symptoms of rhino-conjunctivitis improved. Treg cells increased during the treatment.
	NCT05208528	Allergic Rhinitis	*Bifidobacterium longum* ES1 (daily)	R	-	Determine whether the treatment reduces the symptoms associated with allergic rhinitis by modulating the gut microbiota and enhancing Tregs.
SCFA/Fiber	NCT05152615	Rheumatoid Arthritis	SCFA Dietary Supplement (3 times daily) and methotrexate	R	-	Study whether oral butyrate modifies the gut microbiome and Treg cells.
	NCT05576597	Rheumatoid Arthritis	Sodium Butyrate (daily)	R	-	Evaluate the effects of sodium butyrate supplementation on intestinal inflammation and the changes of percentage of T-cell subtypes, especially Treg numbers.
	NCT04574024	MS	High-Fiber Supplement (daily)	E	-	Analyze the effect of a high-fiber supplement on the growth of SCFA-producing intestinal bacteria and development of Tregs.
Vitamin A	NCT01225289	MS	Vitamin A (daily)	C	[140]	Vitamin A supplementation increased TGF-β and FoxP3 gene expression and may modify the skewing of naive CD4^+^ T towards a Th2/Treg response.
	NCT01668615	Chronic immune thrombocytopenia	All-trans-retinoid acid (3 times a day)	C	[141]	Therapy enhanced the percentage of Treg cells, Foxp3 expression and IL-10 production.
Vitamin D	NCT04472481	Active Rheumatoid Arthritis	Ergocalciferol (Vitamin D2) (weekly)	C	[142]	The percentage of Treg cells increased and was associated with a reduction in the DAS-28 score.
	NCT01390480	T1D	Cholecalciferol (Vitamin D3)(Daily)	C	[143]	Suppressive function of Tregs in patients with T1D improved. The cholecalciferol group required less insulin doses.
	NCT00940719	MS	Vitamin D3 (daily)	C	[144]	Vitamin D3 increases the proportion of IL-10^+^CD4^+^ T cells and decreases the ratio of Th1/Th2 cells.
	IRCT2015010910891N2	MS	High-dose vitamin D3 (every 5 days)	C	[145]	Serum IL-10 levels increased significantly while calcium levels remained stable.
	-	SLE	Vitamin D (daily)	C	[146]	Treg/Th17 ratio increased and decreased SLE DAI scores.
Low dose IL-2 (Ld-IL2)	NCT01353833	T1D	Ld-IL2 (Aldes-leukin) (Daily)	C	[131]	Increase the proportion of Treg cells.
	NCT01988506	Autoimmune inflammatory diseases	Ld-IL2 (daily for 5 days, fortnightly for 6 months)	C	[135]	The Ld-IL2 treatment selectively activated and expanded Tregs and is safe in different diseases and its associated treatments.
	NCT00539695	GVHD	Ld-IL2 (three times weekly)	C	[147]	Ld-IL2 treatment increased Treg cell numbers and may be associated with a lower incidence of viral infections and GVHD.
	NCT02467504	Rheumatoid Arthritis	Ld-IL2 (3 cycles of 14 days, every other day) and Methotrexate	C	[133]	The ratio of Treg/Th17 cells increased in the Ld-IL2 plus Methotrexate group. Inflammatory cytokines were reduced.

C: Completed, R: Recruiting, E: Enrolling.

## 6. Concluding Remarks and Future Perspectives

Autoimmune and chronic-inflammatory diseases, including allergy, have become a major public health concern in developed countries. These diseases are characterized by aberrant and/or excessive immune responses and are treated with immunosuppressive or anti-inflammatory agents to alleviate symptoms. Thereby, there are great expectations on treatments that are aimed to enhance Treg cell numbers and activity, as they have the potential to be curative by restoring immune tolerance and homeostasis. Among these treatments, the adoptive transfer of Tregs is the most direct and advanced approach and is likely to be available in the clinic for the treatment of the aforementioned disorders. Moreover, this approach will likely adopt innovations to overcome shortcomings such as Treg plasticity and lack of specificity. For example, CRISPR/Cas technology could be exploited to stabilize FoxP3 expression in Tregs ex vivo prior to their autologous transference [148,149]. Similarly, the use of antigen-specific engineered Tregs such as CAR-Tregs can increase the activity of Treg cells at specific sites as shown in various murine models of encephalomyelitis [150,151], colitis [152] and arthritis [153,154]. A detailed description of the latest developments in Treg therapy involving engineered antigen-specific Tregs, CAR-Tregs and chimeric Tregs can be found in an excellent review by Eggenhuizen et al. [15]. In this review, the authors also include a relevant section depicting the use tolerogenic DCs in adoptive cell transfer to promote tolerance. Moreover, the authors also describe additional non-cell-based approaches that have been shown to enhance Tregs such as the use of TNF receptor 2 (TNFR2) agonists, the anti-CD20 antibody rituximab and drugs including rapamycin. In comparison, we put a special focus on mucosal Tregs, their ontology and on how to enhance their numbers and activity using microbiota species and products, and micronutrients such as vitamins A and D. These products are readily available and have a great therapeutic potential in autoimmune and chronic-inflammatory diseases. However, additional clinical trials are required to prove the efficacy of these treatments.

## Figures and Tables

**Figure 1 ijms-24-07797-f001:**
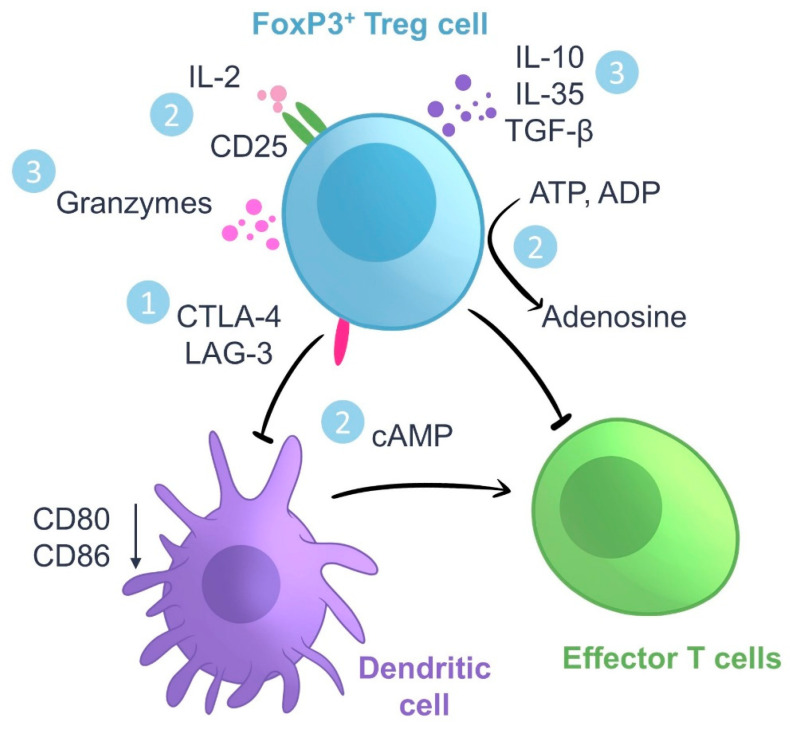
Regulatory T cell (Treg) immunosuppression. FoxP3^+^ Tregs suppress immune responses via three main mechanisms: (1) contact-dependent suppression, (2) metabolic disruption of conventional effector T cells, and (3) the secretion of inhibitory cytokines such as IL-10, IL-35 and TFG-β, which inhibit both conventional T cells and dendritic cells (DCs).

**Figure 2 ijms-24-07797-f002:**
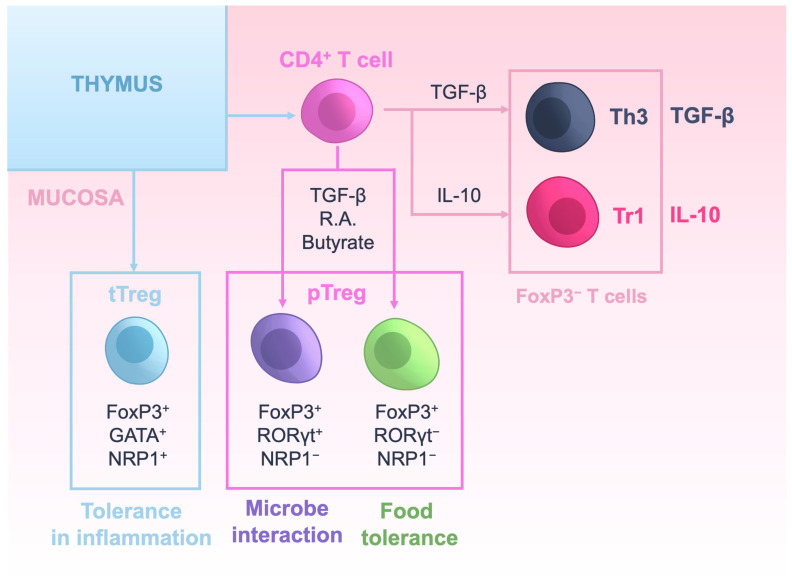
Regulatory T cells in the gut mucosa. In the mucosa, FoxP3^+^ Tregs can be tTregs that differentiate in the thymus or pTregs that differentiate locally from uncommitted CD4^+^ T cells. Two major subsets of pTregs have also been identified based on the expression of RORγt. RORγt^+^ FoxP3^+^ Tregs are important for maintaining tolerance to the microbiota and RORγt^–^FoxP3^+^ Tregs are important for maintaining tolerance to food. Tr1 and Th3 regulatory T cells lacking FoxP3 expression can also be found in the mucosa. Tr1 and Th3 cells secrete IL-10 and TGF-β, respectively, and are induced by classical FoxP3^+^ Tregs and mucosal DCs.

**Figure 3 ijms-24-07797-f003:**
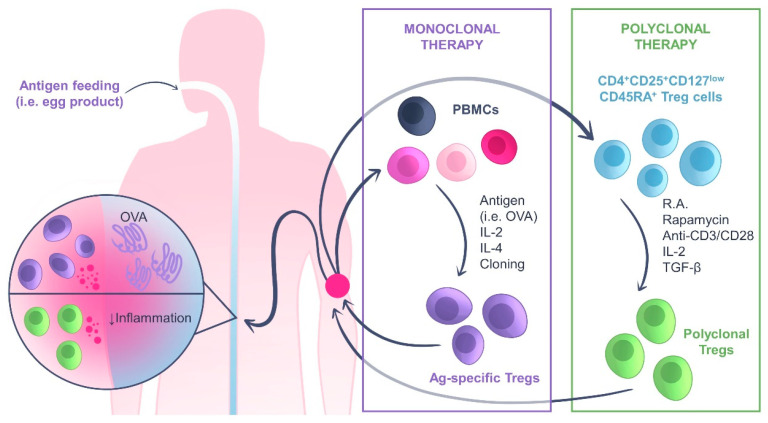
Adoptive Treg cell transfer therapy in IBD. Two approaches can be followed to treat IBD with Tregs: monoclonal or polyclonal therapy. In the monoclonal therapy, PBMCs are isolated from the patient and antigen-specific Treg cells are cloned after expansion with IL-2, IL-4 and the antigen (e.g., OVA). The expanded cells are reinfused into the patient. The patient is fed with the antigen (e.g., egg products) to activate antigen-specific Tregs and suppress excessive inflammation in the gut. In the polyclonal therapy, CD4^+^CD25^+^CD127^low^CD45RA^+^ Treg cells are directly isolated from the patient and expanded with different stimuli (retinoic acid, rapamycin, anti-CD3/CD28, IL-2 and TGF-β). This results in a polyclonal population of Tregs that can be reinfused in the patient to suppress inflammation.

**Figure 4 ijms-24-07797-f004:**
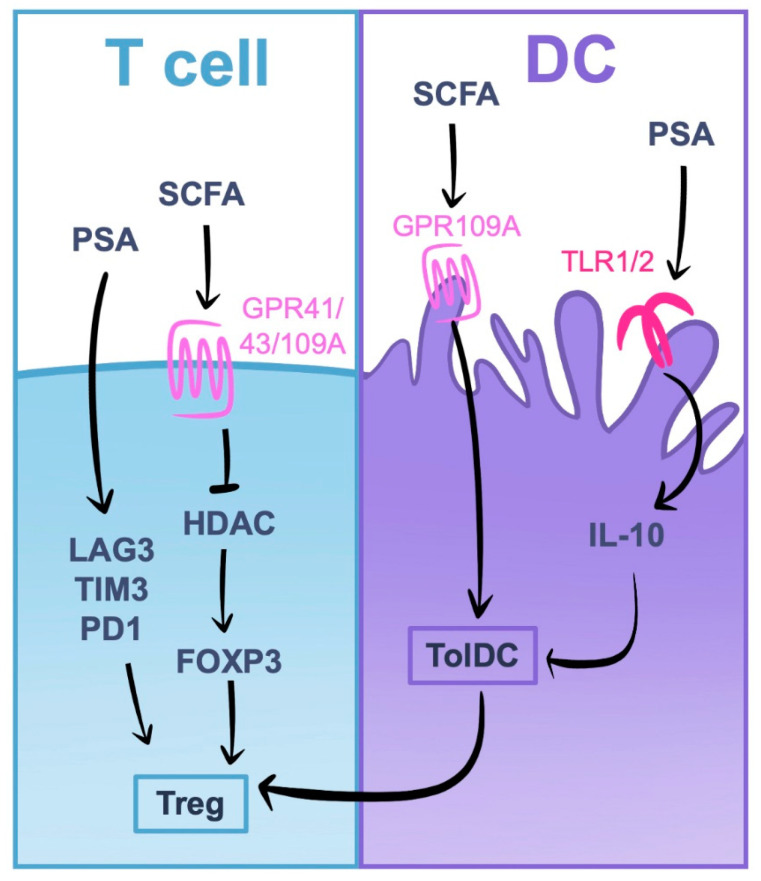
Induction of Treg cells by the microbiota. Short chain fatty acids (SCFAs) produced by the microbiota bound to receptors GPR41, 43 and 109A, inhibiting histone deacetylases (HDAC) and allowing FoxP3 expression and Treg induction. In DCs, SCFAs induce a tolerogenic phenotype (TolDC) that also promotes Treg differentiation. Polysaccharide A (PSA) can also enhance Treg function, as it induces the expression of LAG3, TIM3 and PD1. In addition, PSA induces the transformation of DCs into TolDCs via TLR1/2 and Dectin-1 signaling.

**Table 1 ijms-24-07797-t001:** IBD clinical trials using Adoptive Treg cell transfer.

IBD	Phase	Source of the Cells	Study ID	Status
CD	I/IIa	Antigen-specific autologous expanded Ova-Tregs	Eudract, Number: 2006-004712-44	Completed
	IIb	Antigen-specific autologous expanded Ova-Tregs (Ovasave)	NCT02327221	Terminated
	I/II	Expanded autologous CD4^+^CD25^+^CD127^low^CD45RA^+^ Tregs (TR004 drug)	NCT03185000	Recruiting
UC	I	Autologous ex vivo expanded CD25^+^ Tregs expanded in the presence of rapamycin, IL-2 and CD3/CD28 beads	NCT04691232	Recruiting

## Data Availability

Not applicable.

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
