# Peer review of "Enhancing Regulatory T Cells to Treat Inflammatory and Autoimmune Diseases"

_ijms, 2023, doi:10.3390/ijms24097797_

Round 1

Reviewer 1 Report

This review article discusses how to enhance the regulatory T cells to treat inflammatory and autoim- 2 mune diseases. It is very interesting and informative.

- Authors should take care of English; it needs improvement.

- Updation with 2023 citations is required.

- Follow the journal style in reference format.

- Authors should take care of English; it needs improvement.

Reviewer 2 Report

The manuscript addresses the potential treatment of autoimmune diseases with Treg cells. The design of the review includes an introduction to Treg cells, their ontogeny and mechanisms of action, as well as their adoptive transfer as therapy. It also presents various relevant clinical trials of transfer to Treg cells in ulcerative colitis and Crohn's disease.

1. In Table 2 there is duplication of references 1-13, which refer to results articles linked to clinical trials. The authors must correct this duplicity so that the citation is continuous.

2. The authors cite the work of Eggenhuizen et al, which is a review titled "Treg Enhancing Therapies to Treat Autoimmune Diseases." This work is cited by the authors very briefly as follows: "Treg cells are essential not only to control misguided immune responses and to maintain self-tolerance but also to avoid excessive immune reactions". However, Eggenhuizen's work is a review of the potential of Treg cells in autoimmune diseases, which includes low-dose IL-2, Treg therapy, polyclonal Treg therapy, engineered antigen-specific Treg therapy, chimeric Treg therapy. The work of Eggenhuizen et al also describes multiple clinical trials, including those cited in this manuscript. In the reviewer's opinion, the authors of this manuscript do not value the work of Eggenhuizen et al by citing it so concisely. The work of Eggenhuizen et al is a direct antecedent of this manuscript, for which it is necessary to discuss this work in the conclusions and perspectives section where the authors establish the similarities and differences of their work with respect to the work of Eggenhuizen et al.

Reviewer 3 Report

Following are my comments for the paper:

1) There is a typo in line 139

2) Nicely described T cell maturation and classification of Tregs in section 2 and their mechanism of action with beautifully described figure 1

3) Impressive summary of ongoing clinical trials of adoptive TregCT in section 4

4) There is a typo with dose on lines 250, 273 and 410. It should be 10e6 or 1*10e6

5) Nicely summarized 3 alternative approaches to adaptive Treg CT in section 5
